# Egg Allergy in Children and Weaning Diet

**DOI:** 10.3390/nu14081540

**Published:** 2022-04-07

**Authors:** Carlo Caffarelli, Arianna Giannetti, Arianna Rossi, Giampaolo Ricci

**Affiliations:** 1Clinica Pediatrica Department of Medicine and Surgery, University of Parma, 43126 Parma, Italy; ary.rossi4@gmail.com; 2Pediatric Unit, Istituto Ricovero e Cura a Carattere Scientifico, Azienda Ospedaliero-Universitaria di Bologna, 40138 Bologna, Italy; arianna.giannetti@gmail.com; 3Department of Medical and Surgical Sciences (DIMEC), University of Bologna, 40138 Bologna, Italy; giampaolo.ricci@unibo.it

**Keywords:** weaning, egg, food allergy, dietetic interventions, prevention, infant, egg allergy

## Abstract

Eggs are a fundamental food in the human diet, and together with cow’s milk, they are the most common food allergen. This work highlights the main nutritional characteristics of eggs to show how their absence from a child’s diet can constitute a serious deficiency. We then analyze the risk factors that facilitate the onset of egg allergy. The third part of the paper reports possible interventions to lower the appearance of food allergy that have been occurred in trials. The last part of the paper is a synthesis of this research study that has been taken from several of the latest guidelines or from position papers.

## 1. Introduction

Hen’s egg (HE) allergy is one of the most frequent food allergies in Western countries, and its rate is increasing [1]. It has thus far been estimated that HE allergy affects between 1.6% and 10.1% [2] and as many as 9.5% of children [3,4,5]. The onset most often occurs before the first birthday [6]. The prevalence and age of spontaneous oral tolerance acquisition in childhood vary among studies because of differences in populations, settings and the means for ascertaining the diagnosis. It has generally been found that HE allergy has a prevalence of 1.2–2.9% at 2 years of age [7,8,9,10,11]. Natural resolution has been estimated in up to 80% of cases at 3 years of age [12] and in 38–90% of children within 5–6 years of age [3,13,14,15,16,17,18]. Clinical HE hypersensitivity is lost by 60% of children between 6 and 12 years of age [2]. Tolerance to baked egg is more quickly achieved. Baked HE allergy is outgrown by 94% of patients at 12 years of age [17], while that to otherwise cooked egg is outgrown by 76% at 12 years of age and by 95% at age 18 [13]. The burden of HE allergy is heavy as it can trigger anaphylactic reactions or severe chronic gastroenteropathy. An elimination diet can provoke nutritional imbalance and increase the risk for poor growth. Furthermore, HE allergy reduces the quality of life by limiting the social activities of children and parents, regardless of symptom severity [19,20]. It is also associated with bullying [21], depression, anxiety, attention/deficit hyperactivity disorder [22] and higher healthcare costs [23,24,25]. In view of these issues, various approaches for preventing HE development in children have been proposed. The current review focuses on prevention, especially by dietary interventions in a child’s first year. The paper is based on evidence from the literature, on the recommendations of scientific societies and on clinical experience.

## 2. Eggs as Nutrients

HE is a basic part of the pediatric diet. This makes it important to know its components in order to properly manage an egg-free diet. In addition to energy, fatty acid and protein intake, HE avoidance mainly reduces the intake of B vitamins and vitamin D [26]. HE is composed of three different parts: 63% egg white, 27.5% egg yolk and 9.5% eggshell and membranes. The basic composition of HE may have some minor changes in its nutritional composition due to production methods. A medium HE weighs 44 g and can provide 62.5 calories while a large HE provides 77 calories. The main components consist of water (76%), proteins (12%) and lipids (9.51%). In addition, it also contains low amounts of carbohydrates (0.72%), minerals and vitamins (Table 1) [27].

Egg white is the principal HE component responsible for allergic reactions (Table 2), though alegg yolk protein may also be responsible. It is difficult to separate the white from the yolk, however, without traces of egg white protein contaminating the yolk.

HEs contain vitamins, minerals, high-quality protein with a perfect amino acid profile, and good fats and various other lesser-known nutrients. Almost all the nutrients are contained in the yolk. However, 60% of the high-quality protein in eggs can be found in the egg white. The main elements of egg white are water and proteins (mainly ovalbumin, ovotransferrin and ovomucoid). Egg white consists primarily of about 90% water into which about 10% proteins are dissolved. Unlike the yolk, which is high in lipids, egg white contains almost no fat and less than 1% of carbohydrate [28]. Egg yolk is a homogeneous emulsion, containing proteins and lipids. Egg yolk proteins mainly consist of livetins (38%, α-, β- and γ-forms), lipovitellins (36%, α- and β-forms), low-density lipoproteins LDL (17%) and phosvitin (8–9%). Among these egg yolk proteins, special attention needs to be paid to γ -livetin (also called γ-globulin or immunoglobulin Y) and phosvitin [29]. Immunoglobulin Y is the main antibody derived from chicken egg, and it is popular for the control of enteric infections of either bacterial or viral origin. Phosvitin has the highest phosphorus content and has high antioxidant activity associated with a high metal binding capacity. Phosvitin has the capacity to bind much more Fe than what is naturally found in egg yolk. Egg yolk lipids comprise 65% neutral lipids, 30% phospholipids and 4% cholesterol [30]. Phospholipids represent the most significant component of egg lipids, and they consist of 78% phosphatidylcholine (PC), 17% phosphatidylethanolamine (PE), 2.5% sphingomyelin (SM), and 0.5% phosphatidylinositol, phosphatidylserine and lysophosphatidylcholine [31]. Egg yolk is a better source for phospholipids than oil seeds. PC plays a vital role in the functions of nerve cells, and it is a key component of biological membranes. Therefore, PC separated from egg yolk can be used in infant formula to improve brain development. PC can also be used as a dietary supplement for choline, an important nutrient for prenatal and postnatal infants whose nervous system is developing quickly [32]. Among egg-derived phospholipids, the most critical ones are in egg yolk. Choline is an essential nutrient for humans, and a dietary deficiency of choline in humans causes fatty liver [33,34]. PE is mainly found in the white matter of the brain and in the neural tissues of the spinal cord. High amounts of PE can induce relaxation and overall improvements in cognitive function. PE also functions as a critical anticoagulant at the luminal endothelial surface of the aortic flow dividers, the ascending aorta and the outer curvature of the aortic arch [35]. SM is an essential component of central nervous system myelin sheaths. It also affects the viability of brain cells and signal transduction in T-cell activation (Table 3) [36,37].

## 3. Risk Factors for Egg Allergy

Infants with a family history of allergy and/or atopic diseases [38,39], but not maternal asthma, allergic rhinitis [40], atopic eczema and sensitization to HE, are at high risk for developing HE clinical allergy. Atopic eczema is frequently related to food allergies, especially to HE, cow’s milk allergy and, in countries with a high consumption, peanut allergy. An IgE-mediated HE allergy has been found in 42% of eczematous children [41]. In population studies, eczematous infants were 5.8 times more likely to be affected by HE allergy than healthy children [42] and 6.18 times more likely to be sensitized against certain foods [43]. The relative risk of developing specific IgE antibodies to HE is 14 months in infants with atopic eczema. However, it should be noted that a third of infants with HE sensitization did not suffer from atopic dermatitis [44]. Atopic eczema seems to act as a canary in the coal mine, since it can be diagnosed 3.5 months before egg allergy, and the severity of the eczema is correlated with a higher probability of HE allergy [45]. The likelihood of HE allergy is higher in males with early-onset atopic eczema [46]. Eczematous infants or infants with cow’s milk allergy who have a positive skin-prick test (serum specific IgE to HE) and who have never ingested HE have a higher chance of developing a reaction to HE upon the first ingestion. In such infants, the rate of clinical hypersensitivity reaction to HE on the first known exposure varies from 42% to 72% [47,48,49]. Palmer et al. [50] showed that 20% of infants with moderate to severe atopic eczema reacted on the first HE intake, and 36% had positive serum IgE to HE at 4 months of age. Patch tests for HE are of little help for identifying children with egg allergies [51]. These findings suggest that IgE relative to HE tests should be performed before the first HE intake in infants with atopic eczema or food allergy [52]. When IgE tests have a positive result, the initial HE introduction needs medical supervision [53]. Moreover, population studies have shown that, irrespective of suffering from atopic eczema, most infants are sensitized to HE before weaning at 4–6 months of age [44]. In total, 4–6 out of 18–20 infants with a positive skin prick test relative to eggs before its introduction into the diet have a positive HE challenge [54]. Overall, these results indicate that sensitization to HE likely occurs following allergen exposure in utero and, after birth, through breast milk, topical application to the skin or inhalation, but not after weaning. Growing data show that allergen exposure through the skin provokes food allergy, while oral exposure induces allergens specific tolerance [55]. It has been hypothesized that the association between HE allergy and atopic eczema may be explained by the facilitated passage of allergens through inflamed skin, which promotes the activation of a Th2 dominant response [56] and leads to allergic sensitization [57]. In agreement with this hypothesis, the epicutaneous application of ovalbumin on damaged skin induces sensitization and anaphylactic reactions after gastric challenge in mice [58]. It is noteworthy that the regular use of moisturizers involves a risk of food allergy to infants irrespective of eczema [59]. Moisturizers increase transepidermal water loss and might favor sensitization by increased percutaneous penetration of food allergens or the disruption of the skin barriers leading to inflammation. In animals with filaggrin loss-of-function mutations, contact with HE protein on the undamaged skin barrier can induce local inflammation and transcutaneous sensitization [58]. However, it is unclear whether filaggrin mutations are linked with food hypersensitivity in infants without atopic eczema [60,61]. Taking into consideration that the exposure of damaged skin to HE induces allergic sensitization, the treatment of eczema is a first-line preventive measure. Accordingly, topical steroids reduce food allergy development by inducing remission of atopic eczema [62]. Additional environmental factors have been associated with HE allergy [63]. It has been hypothesized that when the immune system lacks microbial exposure in infancy, the normal maturity of the immune system against infections is not reached and allergy develops. HE allergy, therefore, more easily occurs in children treated with antibiotics during the first week of life [45]. The absence of contact with vaginal and perianal maternal flora during childbirth may select a different child microbiome that is believed to favor the Th2 phenotype and food allergy. Several studies have not observed the association of HE allergy with caesarean delivery [64,65,66]; however, when it was detected [44,67], it was unclear whether it was due to a publication bias [68]. Traffic-related air pollution but not second-hand tobacco smoke exposure [69] has been found to increase the risk for HE sensitization [70].

## 4. Dietary Interventions

Current evidence suggests that in infancy it may be possible to take advantage of a time window to introduce the main food allergens for inducing oral tolerance (“window of opportunity”) [44]. An early and continuous oral intake of food protein may induce long-lasting immune tolerance [63], with systemic immune unresponsiveness to ingested allergens, through the GALT (Gut Associated Lymphoid Tissue). The ability of the GALT to ensure a response against pathogens and to suppress that against commensal bacteria and foods is favored by the intestinal microbiota during weaning [71]. The timing of HE introduction into the diet may, therefore, play a part in the development of HE allergy. On the one hand, there is no evidence that avoidance of HE during pregnancy or breast-feeding, or postponing introduction to the weaning diet, prevents clinical HE allergy [72,73,74,75]. On the other hand, there are some observations that the early and habitual consumption of HE after birth can decrease the occurrence of HE allergy. The Australian HealthNuts study showed that HE intake between 4 and 6 months of age was related to a decreased chance of HE allergy, compared to delaying HE exposure to between 10 and 12 months of age or beyond 12 months of age [76]. Furthermore, Lai et al. [77] found an increased expansion of ovalbumin-specific T regulatory cells and no HE allergy in infants eating HE between 5 and 10 months of age, in contrast to those who had been introduced to HE after 10 months. In infants with bronchiolitis, 82% of 770 participants were introduced to HE in their diet by age twelve months. The cumulative incidence of likely HE allergy by three years of age was 0.2% among children with HE ingestion before 12 months and 2.2% among children who were introduced to HE later [78]. Discordant results have been found, however, in six randomized placebo-controlled trials conducted for assessing the efficacy of an early HE introduction into the diet when complementary feeding starts to prevent HE allergy. In infants with atopic eczema, Natsume et al. [79] administered 50 mg/day (about 1/160th of one HE) of cooked lyophilized HE or placebo from 6 to 9 months and then 250 mg/day or placebo until 12 months. At 1 year of age, open HE oral challenge confirmed HE allergy in 8% of infants in the active group and in 38% of controls (RR (95% CI): 0.22 (0.09–0.54); *p* = 0.0001). However, two factors should be considered. First, infants were treated with topical steroids to maintain remission. This may have reduced the sensitization rate [62]. Second, the food challenge was not conducted in 26 members (17%) of the study population, and ITT analysis was not performed. Perkin et al. [54] enrolled 1303 breast-fed infants aged 3 months. A group that was introduced earlier to six allergenic foods (including boiled HE) between three and six months of age was compared to infants with standard weaning. HE allergy was observed in 1.4% of infants in the early-introduction group and in 5.5% of infants in the standard-introduction group between 1 year and 3 years of age (*p* = 0.009). They also found that the intake of cooked (boiled) HE before 6 months predisposed subjects to the occurrence of food protein-induced enterocolitis syndrome (FPIES) to HE. In contrast, Bellach et al. [44] found that the introduction of pasteurized HE that is considered equivalent to raw egg [80] before 6 months of age did not decrease HE allergy onset. They fed a general population 2.5 g (about one-third of one egg) 3 times per week of pasteurized white HE or with a rice placebo from between 4 and 6 to 12 months. At 1 year, the HE oral challenge had a positive result in 2.1% of the active group and in 0.6% of the placebo group, respectively (RR (95% CI): 3.30 (0.35–31.32); *p* = 5.35). In the Solid Timing for Allergy Research (STAR) trial [50], 86 infants from 4–6 months to 8 months with moderate to severe atopic eczema had 0.9 g (about one-sixth of one egg) daily of HE protein, whole pasteurized raw HE or rice. At one year of age, HE allergy was diagnosed in 33% of treated children and in 51% of controls (RR (95% CI): 0.65 (0.38–1.1); *p* = 0.11). In the Starting Time of Egg Protein (STEP) trial [81], infants who had never eaten HE and without allergic diseases, but had mothers suffering from atopic conditions, had 0.9 g of pasteurized raw egg per day (about one half a HE per week) or rice (placebo) from 4–6 months up to 10 months of age. At 12 months, HE challenges resulted positive in 7% of children in the active group and in 10.3% of the control group (RR (95% CI) 0.75 (0.48–1.17); *p* = 0.20). In the Beating Egg Allergy Trial (BEAT) [82], infants with one or birth parents presenting atopic disorder were introduced to 350 mg of pasteurized whole HE from 4 to 8 months, after successful first weaning. At 1 year of age, 6.2% of subjects in the active group had HE allergy, compared to 10.5% in the placebo group (*p* > 0.05). Early exposure to raw HE was further linked with anaphylaxis [44,50]. It seems that a complementary feeding that includes the daily intake of a very small amount of cooked HE probably decreases the onset of HE allergy in the first year of life.

## 5. What about Guidelines

With a nod to recent trials, recommendations on the timing of introduction of HE in infancy have been released (Table 4). The European Food Safety Authority (EFSA) [83] suggests HE introduction between 4 and 6 months. The Australasian Society of Clinical Immunology and Allergy (ASCIA) [84] recommends that all infants should be given allergenic solid foods, including cooked HE, in the first year of life. It states that there is moderate evidence that introducing cooked HE (raw HE is not recommended) into the diet before 8 months of age in infants with a family history of allergy could reduce the risk of developing HE allergy. Both the European Academy of Allergy and Clinical Immunology (EAACI) [38] and the American Academy of Allergy, Asthma, and Immunology/Canadian Society of Allergy and Clinical Immunology (AAAAI/CSACI) guidelines [85] suggest introducing well-cooked HE but not raw egg or uncooked pasteurized HE into the infant diet from 4 to 6 months of life as part of complementary feeding to prevent HE allergy in infants. In families with infants at general and increased risk, EAACI guidelines [38] suggest introducing about half a well-cooked, small HE twice a week as part of complementary feeding from 4 to 6 months of age. AAAA/CSACI guidelines [85] suggest the introduction of HE or egg-containing products to all infants, irrespective of their relative risk of developing allergy, between 4 and 6 months of life using only cooked forms of egg and avoiding any raw, pasteurized egg-containing products. The British Society for Allergy and Clinical Immunology (BSACI) [86] recommends starting complementary foods, including HE, alongside breastfeeding, when the infant is ready around 6 months of age (but not before 4 months) since delaying HE from the infant diet does not prevent allergy. Infants with atopic eczema should be introduced to solids, including HE, between 4 and 6 months, when they are ready. A small amount (1 teaspoon) of well-cooked HE egg should initially be given; then, the amount can be gradually increased to a full dose. The Deutschen Gesellschaft für Allergologie und Klinische Immunologie (DGAKI) guideline [87] advises the introduction of complementary foods and HE from 4 to 6 months of age. Baked or hard-boiled HE should be regularly given, while raw eggs (including scrambled and soft-boiled eggs) are not recommended.

## 6. Conclusions

HE is a fundamental food for the development of the child. It contains not only a complete protein profile, which includes all the essential amino acids for growth, but also includes antioxidants, microelements and phospholipids, which have a key role in the maturation of the nervous system. HE allergy, which implies a HE-free diet, must therefore be carefully diagnosed to avoid generalizations and the transformation of a simple sensitization into a clinical diagnosis. In children with HE allergy, it is necessary to periodically check whether they outgrow it. When tolerance to HE is not achieved, oral immunotherapy for HE can be performed. The method of preventing HE allergy should be carefully assessed. The debate on the timing of HE introduction in weaning has led to a change in positions over the last decade. Several studies to which the guidelines have been subsequently adapted identify the opportunity of not delaying the introduction of HE at the time of weaning. HE introduction should be avoided before the fourth month of age. Moreover, there is only one study supporting the preventive effect of introducing cooked HE after the sixth month of age in infants with atopic eczema. Looking at the studies taken together, a delayed introduction has no preventive benefit and may negatively influence the growth and psychological wellbeing of children and their families. In agreement with the most recent guidelines (Table 3), we suggest that HE or HE-containing products should be a regular part of the diet from around 6 months of age, and they should not be introduced earlier than 4 months of age. This can be applied to all infants, even those without atopic eczema or when an atopic family history is lacking, regardless of the relative risk of developing HE allergy. At weaning, well-cooked HE, such as baked and boiled HE, should be given starting with small amounts, while any raw, pasteurized egg-containing products should be avoided to reduce the development of HE clinical hypersensitivity. In infants with mainly moderate to severe atopic eczema or food allergy, it is advisable to carry out the HE skin-prick test before the introduction of HE. When the HE skin-prick test is positive, performing the first HE intake under medical supervision should be considered. Finally, we must not forget that starting weaning, as suggested by EFSA [83], depends on the acquisition of neuromotor skills (such as head and trunk control which allow improved movements of jaw, lip, and tongue) and anatomical changes in the mouth to protect infants from aspiration and choking.

## Figures and Tables

**Table 1 nutrients-14-01540-t001:** Main nutrient contents in one boiled or poached egg weighing 44 g according to the United States Department of Agriculture [27].

Protein 5.5 g
Total fat: 4.2 g, of which 1.4 g are saturated
Cholesterol: 162 mg
Sodium: 189 mg
Phosphorus: 86.7 mg
Potassium: 60.3 mg
Calcium: 24.6 mg
Magnesium 5.3 mg
Iron: 0.8 mg
Lutein and zeaxanthin: 220 μg
Folate: 15.4 μg
Selenium: 13.4 μg

**Table 2 nutrients-14-01540-t002:** Differences in allergenic proprieties between egg yolk and egg white proteins.

Egg White Protein	Egg Yolk Protein
ovomucoid (Gal d 1)	Phosvitin
ovalbumin (Gal d 2)	α-livetin (Gal d 5)
ovotransferrin or conalbumin (Gal d 3)	apovitellenins I
egg lysozyme (Gal d 4)	apovitellenins VI (or apoprotein B)
ovomucin	

**Table 3 nutrients-14-01540-t003:** Functions and benefits of eggs in a child’s diet (modified from [37]).

Contain high-quality proteins with all nine essential amino acids
Improves cholesterol profile increasing HDL and does not raise the risk of heart disease
A good source of omega-3s. Omega-3s play an important role in the way cell membranes work, from heart and brain health to protecting the eyes
Contain choline, a nutrient that contributes mainly to healthy brain development. Choline is required to synthesize the neurotransmitter acetylcholine and is also a component of cell membranes.
Are a great source of Vitamin D.
Have an antioxidant effect: the presence of the carotenoids lutein and zeaxanthin improves the pigment density in the retina. Vitamin A, vitamin E and selenium also help.
Help with weight management because they are relatively low in calories. The high satiety levels of eggs leads to greater feelings of satisfaction, less hunger and a lowered desire to eat later in the day.

**Table 4 nutrients-14-01540-t004:** Guidelines on dietary interventions for preventing egg allergy.

Guideline	Year	Healthy Children	Children with Atopic Eczema	Children with Food Allergy	Family History for Food Allergy
EFSA [83]	2019	Egg introduction at 3–4 months of age compared with 6 months of age may reduce the risk of developing egg allergy (low to moderate confidence in the evidence). There were some anaphylactic reactions associated with the consumption of raw egg but not with cooked egg.	Egg introduction between 4 and 6 months of age may be associated with a lower risk of developing egg allergy at 1 year of age.	Egg introduction between 4 and 6 months of age may be associated with a lower risk of developing egg allergy at 1 year of age	Egg introduction between 4 and 6 months of age may be associated with a lower risk of developing egg allergy at 1 year of age
ASCIA [84]	2020	Between four and six months, start to introduce a variety of solid foods, while continuing breastfeeding. Introduce allergenic solid foods, including cooked egg products, in the first year of life.	As in the healthy child.	As in the healthy child.	Introducing cooked egg (raw egg is not recommended) before 8 months of age, where there is a family history of allergy, can reduce the risk of developing egg allergy.
EAACI [38]	2021	Introducing well-cooked hen’s egg, but not raw egg or uncooked pasteurised egg, into the infant diet as part of complementary feeding to prevent egg allergy in infants from 4 to 6 months of life.	Families with infants at general and increased risk to start introducing about half of a well-cooked, small egg twice a week as part of complementary feeding from 4 to 6 months of age.	As in the children with eczema.	NS
AAAAI/CSACI [85]	2021	Egg should be introduced around 6 months of age, but not before 4 months.	Introduce egg or egg-containing products to all infants (only cooked forms of egg and avoid administering any raw, pasteurized egg-containing products) around 6 months of age, though not before 4 months of age.	NS	NS
BSACI [86]	2021	Exclusive breastfeeding until around 6 months of age with complementary foods from around this age.	Introduce egg as a starting weaning food after more traditional weaning foods between 4 and 6 months of age. Give well-cooked egg starting with 1 teaspoon and gradually increase to a full dose.	NS	NS
DGAKI [87]	2021	Exclusive breastfeeding for the first 4–6 months that should continue during introduction of complementary foods. Egg should be introduced at weaning as described for infants with family history.			At weaning, introduction and regular administration of heated-through egg (baked, hard-boiled). It is not recommended to introduce raw egg (including scrambled and soft-boiled eggs).

## Data Availability

Not applicable.

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
