# Peer review of "Egg Allergy in Children and Weaning Diet"

_nutrients, 2022, doi:10.3390/nu14081540_

Round 1

Reviewer 1 Report

The review entitled “Egg Allergy in Children in Weaning Diet” by Caffarelli Carlo, and coworkers is a good revision of Hen´s egg (HE) allergy that provides background information about the disease and summarizes the different existing guidelines on the optimal timing for regular egg intake.

The quality of the information revised makes this manuscript acceptable for publication. However, there are some suggestions we would like to make to the authors in order to improve the paper.

The length of the manuscript, introduction, and references are adequate. The sections on data presentation are generally clear. However, some data seem poorly presented, as in Table 1, the main nutrient contents in one boiled or poached egg. We suggest improving this table by including yolk and white albumen components, as two additional columns, with the information given below in the text.

We have detected editing errors as extra spaces between words that will be easily corrected.

Author Response

The review entitled “Egg Allergy in Children in Weaning Diet” by Caffarelli Carlo, and coworkers is a good revision of Hen´s egg (HE) allergy that provides background information about the disease and summarizes the different existing guidelines on the optimal timing for regular egg intake.The quality of the information revised makes this manuscript acceptable for publication. However, there are some suggestions we would like to make to the authors in order to improve the paper.

Answer. We sincerely thank the reviewer for his/her comment.

The length of the manuscript, introduction, and references are adequate. The sections on data presentation are generally clear. However, some data seem poorly presented, as in Table 1, the main nutrient contents in one boiled or poached egg. We suggest improving this table by including yolk and white albumen components, as two additional columns, with the information given below in the text.

Answer. We are grateful to the reviewer for his/her criticism. It becomes complex to expand table 1 that refers to nutritional properties, as the white egg is made up almost entirely of proteins only. We have added  a new table 2 and a sentence that emphasizes the requested information.

We have detected editing errors as extra spaces between words that will be easily corrected.

Answer. The English language has been revised throughout the paper

Reviewer 2 Report

General comment:

Good revision. However I miss an explanation, even in a specific heading, at the beginning about main allergens and differences between white and yolk allergen content.

The allergenic content of yolk and white egg is very different. Both can be allergenic although usually hen’s egg allergy in children is associated with white egg; yolk allergy is less frequent and usually develops later in life, associated with respiratory allergy to feathers (bird-egg syndrome). Besides, when authors describe the introduction in the diet of egg, boiled, pasteurized etc., they do not explain if they are referring to whole egg or only white egg.

Specific comment:

It would be desirable a list of abbreviators to ease the reading.

Table 2 and 3 should be improved schematizing and summarizing the content.

Author Response

General comment:

Good revision. However I miss an explanation, even in a specific heading, at the beginning about main allergens and differences between white and yolk allergen content. The allergenic content of yolk and white egg is very different. Both can be allergenic although usually hen’s egg allergy in children is associated with white egg; yolk allergy is less frequent and usually develops later in life, associated with respiratory allergy to feathers (bird-egg syndrome).

Answer. We are grateful to the reviewer for his/her criticism. We have added  a new table 2 and a sentence that emphasizes the requested information.

Besides, when authors describe the introduction in the diet of egg, boiled, pasteurized etc., they do not explain if they are referring to whole egg or only white egg.

Answer. According to the reviewer comment, the paper has been corrected.

Specific comment:

It would be desirable a list of abbreviators to ease the reading.

Answer. We have added a list of abbreviations after the keywords.

Table 2 and 3 should be improved schematizing and summarizing the content.

Answer. According to the reviewer’s comment, we have modified  Table 2 and 3.